# Improving Adversarial Robustness via Channel-wise Activation Suppressing

**Yang Bai[1]*  Yuyuan Zeng[2,5]*  Yong Jiang[1,2,5]  Shu-Tao Xia[2,5]†  Xingjun Ma[4]  Yisen Wang[3]†**
[1]Tsinghua Berkeley Shenzhen Institute, Tsinghua University, China
[2]Tsinghua Shenzhen International Graduate School, Tsinghua University, China
[3]Key Lab. of Machine Perception (MoE), School of EECS, Peking University, Beijing, China
[4]School of Information Technology, Deakin University, Geelong, VIC, Australia
[5]PCL Research Center of Networks and Communications, Peng Cheng Laboratory, Shenzhen, China

## Abstract

The study of adversarial examples and their activation has attracted significant attention for secure and robust learning with deep neural networks (DNNs). Different from existing works, in this paper, we highlight two new characteristics of adversarial examples from the channel-wise activation perspective: 1) the activation magnitudes of adversarial examples are higher than that of natural examples; and 2) the channels are activated more uniformly by adversarial examples than natural examples. We find that the state-of-the-art defense adversarial training has addressed the first issue of high activation magnitudes via training on adversarial examples, while the second issue of uniform activation remains. This motivates us to suppress redundant activation from being activated by adversarial perturbations via a Channel-wise Activation Suppressing (CAS) strategy. We show that CAS can train a model that inherently suppresses adversarial activation, and can be easily applied to existing defense methods to further improve their robustness. Our work provides a simple but generic training strategy for robustifying the intermediate layer activation of DNNs. Code is available at https://github.com/bymavis/CAS_ICLR2021.

## 1 Introduction

Deep neural networks (DNNs) have become standard models for solving real-world complex problems, such as image classification (He et al., 2016), speech recognition (Wang et al., 2017), and natural language processing (Devlin et al., 2019). DNNs can approximate extremely complex functions through a series of linear (*e.g.* convolution) and non-linear (*e.g.* ReLU activation) operations. Despite their superb learning capabilities, DNNs have been found to be vulnerable to adversarial examples (or attacks) (Szegedy et al., 2014; Goodfellow et al., 2015), where small perturbations on the input can easily subvert the model's prediction. Adversarial examples can transfer across different models (Liu et al., 2017; Wu et al., 2020a; Wang et al., 2021) and remain destructive even in the physical world (Kurakin et al., 2016; Duan et al., 2020), raising safety concerns in autonomous driving (Eykholt et al., 2018) and medical diagnosis (Ma et al., 2021).

Existing defense methods against adversarial examples include input denoising (Liao et al., 2018; Bai et al., 2019), defensive distillation (Papernot et al., 2016), gradient regularization (Gu & Rigazio, 2014), model compression (Das et al., 2018) and adversarial training (Goodfellow et al., 2015; Madry et al., 2018; Wang et al., 2019), amongst which adversarial training has demonstrated the most reliable robustness (Athalye et al., 2019; Croce & Hein, 2020b). Adversarial training is a data augmentation technique that trains DNNs on adversarial rather than natural examples. In adversarial training, natural examples are augmented (or perturbed) with the worst-case perturbations found within a small $L_p$-norm ball around them. This augmentation has been shown to effectively smooth out the loss landscape around the natural examples, and force the network to focus more on the pixels that are most relevant to the class. Apart from these interpretations, it is still not well understood,

---

*Equal contribution.
†Correspondence to: Yisen Wang (yisen.wang@pku.edu.cn), Shu-Tao Xia (xiast@sz.tsinghua.edu.cn).

from the activation perspective, how small input perturbations accumulate across intermediate layers to subvert the final output, and how adversarial training can help mitigate such an accumulation. The study of intermediate layer activation has thus become crucial for developing more in-depth understanding and robust DNNs.

In this paper, we show that, if studied from a channel-wise perspective, strong connections between certain characteristics of intermediate activation and adversarial robustness can be established. Our channel-wise analysis is motivated by the fact that different convolution filters (or channels) learn different patterns, which when combined together, describe a specific type of object. Here, adversarial examples are investigated from a new perspective of channels in activation. Different from the existing activation works assuming different channels are of equal importance, we focus on the relationship between channels. Intuitively, different channels of an intermediate layer contribute differently to the class prediction, thus have different levels of vulnerabilities (or robustness) to adversarial perturbations. Given an intermediate DNN layer, we first apply global average pooling (GAP) to obtain the channel-wise activation, based on which, we show that the activation magnitudes of adversarial examples are higher than that of natural examples. This means that adversarial perturbations generally have the signal-boosting effect on channels. We also find that the channels are activated more uniformly by adversarial examples than that by natural examples. In other words, some redundant (or low contributing) channels that are not activated by natural examples, yet are activated by adversarial examples. We show that adversarial training can effectively address the high magnitude problem, yet fails to address the uniform channel activation problem, that is, some redundant and low contributing channels are still activated. This to some extent explains why adversarial training works but its performance is not satisfactory.

Therefore, we propose a new training strategy named Channel-wise Activation Suppressing (CAS), which adaptively learns (with an auxiliary classifier) the importance of different channels to class prediction, and leverages the learned channel importance to adjust the channels dynamically. The robustness of existing state-of-the-art adversarial training methods can be consistently improved if applied with our CAS training strategy. Our key contributions are summarized as follows:

- We identify, from a channel-wise activation perspective, two connections between DNN activation and adversarial robustness: 1) the activation of adversarial examples are of higher magnitudes than that of natural examples; and 2) the channels are activated more uniformly by adversarial examples than that by natural examples. Adversarial training only addresses the first issue of high activation magnitudes, yet fails to address the second issue of uniform channel activation.

- We propose a novel training strategy to train robust DNN intermediate layers via Channel-wise Activation Suppressing (CAS). In the training phase, CAS suppresses redundant channels dynamically by reweighting the channels based on their contributions to the class prediction. CAS is a generic intermediate-layer robustification technique that can be applied to any DNNs along with existing defense methods.

- We empirically show that our CAS training strategy can consistently improve the robustness of current state-of-the-art adversarial training methods. It is generic, effective, and can be easily incorporated into many existing defense methods. We also provide a complete analysis on the benefit of channel-wise activation suppressing to adversarial robustness.

## 2 RELATED WORK

**Adversarial Defense.** Many adversarial defense techniques have been proposed since the discovery of adversarial examples. Among them, many were found to have caused obfuscated gradients and can be circumvented by Back Pass Differentiable Approximation (BPDA), Expectation over Transformation (EOT) or Reparameterization (Athalye et al., 2019). Adversarial training (AT) has been demonstrated to be the most effective defense (Madry et al., 2018; Wang et al., 2019; 2020b), which solves the following min-max optimization problem:

$$\min_{\theta} \max_{\boldsymbol{x}' \in \mathcal{B}_\epsilon(\boldsymbol{x})} \mathcal{L}(\mathcal{F}(\boldsymbol{x}', \theta), y), \tag{1}$$

where, $\mathcal{F}$ is a DNN model with parameters $\theta$, $\boldsymbol{x}$ is a natural example with class label $y$, $\boldsymbol{x}'$ is the adversarial example within the $L_p$-norm ball $\mathcal{B}_\epsilon(\boldsymbol{x}) = \{\boldsymbol{x}' : \| \boldsymbol{x}' - \boldsymbol{x} \|_p \le \epsilon\}$ centered at $\boldsymbol{x}$,

$\mathcal{F}(\boldsymbol{x}', \theta)$ is the output of the network, and $\mathcal{L}$ is the classification loss (*e.g.* the cross-entropy loss). The inner maximization problem is dependent on the adversarial examples $\boldsymbol{x}'$ generated within the $\epsilon$-ball, while the outer minimization problem optimizes model parameters under the worst-case perturbations found by the inner maximization. There are other variants of adversarial training with some new objective function or regularization. For example, TRADES (Zhang et al., 2019) optimized a trade-off objective of adversarial robustness and accuracy. MART (Wang et al., 2020c) applied a distinctive emphasis on misclassified versus correctly classified examples. AWP (Wu et al., 2020b) incorporated the regularization on the weight loss landscape. However, apart from these improvements, it is still not well understood how adversarial training can help produce state-of-the-art robustness from the activation perspective.

**Activation Perspective of Adversarial Robustness.** Some previous works have investigated the adversarial robustness from the architecture perspective, such as skip connection (Wu et al., 2020a) and batch normalization (Galloway et al., 2019). As for intermediate activation, Ma et al. (2018) characterized that adversarial activation forms an adversarial subspace that has much higher intrinsic dimensionality. Zhang et al. (2018) certified the robustness of neural networks with different activation functions. Xu et al. (2019) explored the influence of adversarial perturbations on activation from suppression, promotion and balance perspectives. Other works developed new activation operations with the manifold-interpolating data-dependent function (Wang et al., 2020a) and adaptive quantization techniques (Rakin et al., 2018). While these works directly modify the activation functions, there are also works focusing on the activation outputs. For instance, $k$-Winner-Takes-All ($k$WTA) (Xiao et al., 2020) taked the largest $k$ feature values in each activation layer to enhance adversarial robustness. However, this has recently been shown not robust against adaptive attacks (Tramer et al., 2020). Stochastic Activation Pruning (SAP) (Dhillon et al., 2018) taked the randomness and the value of features into consideration. Each activation is chosen with a probability proportional to its absolute value. Adversarial Neural Pruning (ANP) (Madaan & Hwang, 2020) pruned out the features that are vulnerable to adversarial inputs using Bayesian method. Prototype Conformity Loss (PCL) (Mustafa et al., 2019) was proposed to cluster class-wise features and push class centers away from each other. Feature Denoising (FD) (Xie et al., 2019) added denoising layers to the network for sample-wise denoising on feature maps. However, these methods were developed based on observations on the full output (*e.g.* the entire feature or activation map that does not distinguish different channels) of DNN intermediate layers. In contrast, our CAS explores both channel importance and channel correlations, and the suppressing is done with the guidance of the labels.

## 3 CHANNEL-WISE ACTIVATION AND ADVERSARIAL ROBUSTNESS

In this part, we investigate two characteristics of DNN intermediate activation from a channel-wise perspective, and show two empirical connections between channel-wise activation and adversarial robustness. Specifically, we train ResNet-18 (He et al., 2016) and VGG16 (Simonyan & Zisserman, 2014) on CIFAR-10 (Krizhevsky et al., 2009) using both standard training and adversarial training under typical settings. We then apply global average pooling to extract the channel-wise activation from the penultimate layer. We investigate the extracted channel-wise activation of both natural and adversarial examples from two perspectives: 1) the magnitude of the activation, and 2) the activation frequency of the channels.

**Channel-wise Activation Magnitude.** Figure 1 illustrates the averaged activation magnitudes for both natural test examples and the corresponding adversarial examples crafted by PGD-20 attack (Madry et al., 2018). For standard models (trained on natural examples), the activation magnitudes of adversarial examples are generally higher than that of natural examples as shown in Figure 1(a)/1(c). Adversarial perturbation exhibits a clear signal-boosting effect on the channels, which leads to the accumulation of adversarial distortions from the input to the output layer of the network. As shown in Figure 1(b)/1(d), adversarial training can effectively narrow the magnitude gaps between natural and adversarial examples, interestingly by decreasing the activation magnitudes of adversarial examples. This is because adversarial training can restrict the Lipschiz constant of the model at deeper layers (*i.e.* layers close to the output), which reduces the magnitude gaps caused by adversarial perturbations (Finlay et al., 2018; Sinha et al., 2019). Note that network architecture also influences activation magnitudes. Figure 1 show that magnitudes in VGG have much more zero values than those in ResNet, *i.e.* VGG produces more sparse channels than ResNet.

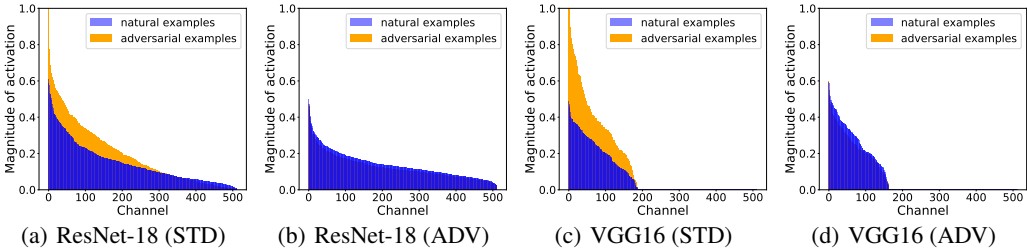

(a) ResNet-18 (STD)     (b) ResNet-18 (ADV)     (c) VGG16 (STD)     (d) VGG16 (ADV)

Figure 1: The magnitudes (y-axis) of channel-wise activation at the penultimate layer (512 channels at x-axis) for both standard ('STD') and adversarially trained ('ADV') models. In each plot, the magnitudes are averaged and displayed separately for natural and adversarial test examples. The 512 channels are sorted in a descending order of the magnitude.

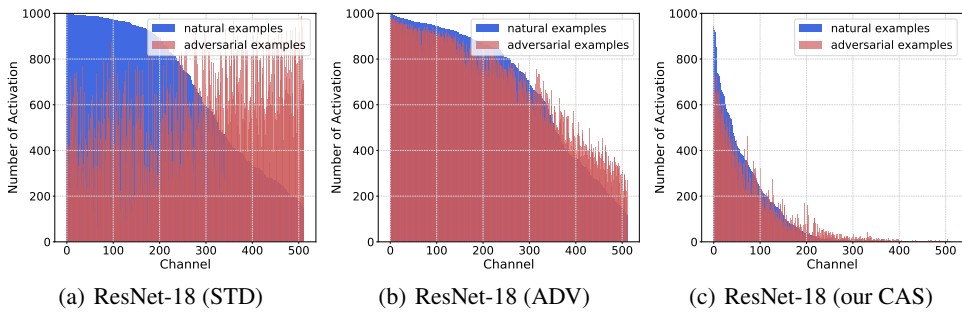

(a) ResNet-18 (STD)     (b) ResNet-18 (ADV)     (c) ResNet-18 (our CAS)

Figure 2: The activation frequency (y-axis) of channel-wise activation at the penultimate layer (512 channels at x-axis) of ResNet-18 trained using (a) standard training ('STD'), (b) adversarial training ('ADV'), and (c) our CAS-based adversarial training ('CAS'). The activation frequencies are counted separately for the natural test examples and their PGD-20 adversarial examples. Channels are sorted in a descending order of activation frequency of natural examples.

**Channel-wise Activation Frequency.** Given a specific class, different convolution filters learn different patterns associated with the class. Similar to the robust *vs.* non-robust feature differentiation in adversarial training (Ilyas et al., 2019), the intermediate filters (or channels) can also be robust or non-robust. Intuitively, for natural examples in the same class, robust channels produce more generic patterns and should be activated more frequently, yet the non-robust ones should be activated less frequently. As such, non-robust channels can cause more variations to the next layer if activated by adversarial perturbations, increasing the vulnerability to adversarial examples. To investigate this, we visualize the activation frequency of the channel-wise activation in Figure 2. Here, we take one specific class (*e.g.* class 0) of CIFAR-10 as an example. A channel is determined as activated if its activation value is larger than a threshold (*e.g.* 1% of the maximum activation value over all 512 channels). We count the activation frequency for each channel by natural examples and their PGD adversarial examples separately on standard or adversarially trained ResNet-18 models, and sort the channels in a descending order according to the activation frequency by natural examples. As can be observed in Figure 2(a), adversarial examples activate the channels more uniformly, and they tend to frequently activate those that are rarely activated by natural examples (*e.g.* the right region in Figure 2(a)). This observation is consistent across different classes. The low frequency channels are non-robust channels, and correspond to those redundant activation that are less important for the class prediction. It can also be observed that adversarial perturbations also inhibit those high frequency channels of natural examples (the left region of Figure 2(a)). Figure 2(b) shows that, by training on adversarial examples, adversarial training can force the channels to be activated in a similar frequency by both natural and adversarial examples. However, there are still a certain proportion of the redundant channels (*e.g.* channels #350 - #500) that are activated by adversarial examples. This motivates us to propose a Channel-wise Activation Suppressing (CAS) training strategy to avoid those redundant channels from being activated by adversarial examples. Figure 2(c) shows the effectiveness of our CAS strategy applied with adversarial training, that is, our CAS can suppress all channels, especially those low frequency ones on natural examples. More visualizations of channel-wise activation frequency can be found in Appendix C.

# 4 PROPOSED CHANNEL-WISE ACTIVATION SUPPRESSING

In this section, we introduce our Channel-wise Activation Suppressing (CAS) training strategy, which dynamically learns and incorporates the channel importance (to the class prediction) into the training phase to train a DNN model that inherently suppresses those less important channels.

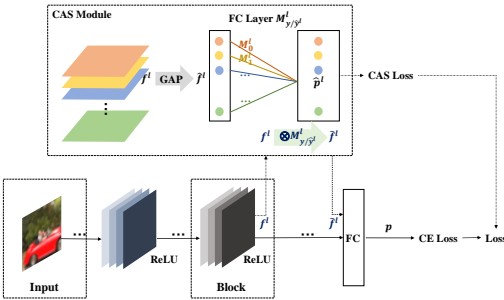

Figure 3: Framework of our proposed Channel-wise Activation Suppressing (CAS).

**Overview.** Figure 3 illustrates our CAS training strategy. The CAS module consists of a global average pooling operation (*i.e.* GAP in the CAS module) to obtain the channel-wise activation, and an auxiliary classifier (*i.e.* FC in the CAS module) to learn the channel importance. The learned channel importance is then multiplied back to the original activation for adjustment, and the adjusted activation are then passed into the next layer for model training. The entire network and the auxiliary classifier are trained simultaneously using a combination of the CAS loss and the CE loss. The CAS module can be attached to any intermediate layer of a DNN.

## 4.1 CAS MODULE

Denote the $l$-th activation layer output of network $\mathcal{F}$ as $\boldsymbol{f}^l \in \mathbb{R}^{H \times W \times K}$, where $H, W, K$ represent the height, width, channel of the activation map, respectively. In CAS module, we first apply the GAP operation on the raw activation $\boldsymbol{f}^l$ to obtain the channel-wise activation $\hat{\boldsymbol{f}}^l \in \mathbb{R}^K$. Formally, for the $k$-th channel,

$$\hat{\boldsymbol{f}}_k^l = \frac{1}{H \times W} \sum_{i=1}^{H} \sum_{j=1}^{W} \boldsymbol{f}_k^l(i, j). \tag{2}$$

The channel-wise activation $\hat{\boldsymbol{f}}^l$ is then passed into the auxiliary classifier to perform multi-class classification with a fully-connected (FC) layer. For $C$ classes, the parameters of the auxiliary classifier can be written as $M^l = [M_1^l, M_2^l, .., M_C^l] \in \mathbb{R}^{K \times C}$, which can identify the importance of each channel to a specific class, and will be applied to reweight the original activation $\boldsymbol{f}^l$ in a channel-wise manner. In the training phase, the ground-truth label $y$ is utilized as the index to determine the channel importance, *i.e.* $M_y^l \in \mathbb{R}^K$. While in the test phase, since the ground-truth label is not available, we simply take the weight component $M_{\hat{y}^l}^l \in \mathbb{R}^K$ that is associated to the predicted class $\hat{y}^l$ as the channel importance (detailed analysis can be found in Section 5.1). The computed channel importance is then applied to reweight the original activation map $\boldsymbol{f}^l$ as follows:

$$\tilde{\boldsymbol{f}}^l = \begin{cases} \boldsymbol{f}^l \otimes M_y^l, & \text{(training phase)} \\ \boldsymbol{f}^l \otimes M_{\hat{y}^l}^l, & \text{(test phase)} \end{cases}, \tag{3}$$

where $\otimes$ represents the channel-wise multiplication. The adjusted $\tilde{\boldsymbol{f}}^l$ will be passed into the next layer via forward propagation. Note that, so far, neither the auxiliary nor the network is trained, just computing the channel importance and adjusting the activation in a channel-wise manner.

## 4.2 MODEL TRAINING

We can insert $S$ Channel-wise Activation Suppressing (CAS) modules into $S$ different intermediate layers of DNNs. The CAS modules can be considered as auxiliary components of the network, and can be trained using standard training or different types of adversarial training. Here, we take the

original adversarial training (Madry et al., 2018) as an example, and define the loss functions to simultaneously train the network and our CAS modules. Each of our CAS modules has a FC layer. Taking one inserted CAS module after the $l$-th activation layer of network $\mathcal{F}$ for examples, the CAS loss function can then be defined as,

$$\mathcal{L}_{\text{CAS}}(\hat{\boldsymbol{p}}^l(\boldsymbol{x}', \theta, M), y) = -\sum_{c=1}^{C} \mathbb{1}\{c = y\} \cdot \log \hat{\boldsymbol{p}}_c^l(\boldsymbol{x}'), \qquad (4)$$

where $C$ is the number of classes, $\hat{\boldsymbol{p}}^l = \text{softmax}(\hat{\boldsymbol{f}}^l M^l) \in \mathbb{R}^C$ is the prediction score of the classifier in CAS module, and $\boldsymbol{x}'$ is the adversarial example used for training. Note that $\mathcal{L}_{\text{CAS}}$ is the cross entropy loss defined on the auxiliary classifier. Similarly, it can also be extended to multiple CAS modules. The overall objective function for adversarial training with our CAS strategy is:

$$\mathcal{L}(\boldsymbol{x}', y; \theta, M) = \mathcal{L}_{\text{CE}}(\boldsymbol{p}(\boldsymbol{x}', \theta), y) + \frac{\beta}{S} \cdot \sum_{s=1}^{S} \mathcal{L}_{\text{CAS}}^s(\hat{\boldsymbol{p}}^s(\boldsymbol{x}', \theta, M), y) \qquad (5)$$

where $\beta$ is a tunable parameter balancing the strength of CAS. Besides the original adversarial training (AT) (Madry et al., 2018), we can also combine CAS with other defense techniques such as TRADES (Zhang et al., 2019) and MART (Wang et al., 2020c). In Appendix B, we summarize the loss functions of the original AT, TRADES, MART, and their combined versions with our CAS. The complete training procedure of our CAS is described in Algorithm 1 in Appendix A.

## 5 EXPERIMENTS

In this section, we first provide a comprehensive understanding of our CAS training strategy, then evaluate its robustness on benchmark datasets against various white-box and black-box attacks.

### 5.1 EMPIRICAL UNDERSTANDING OF CAS

In this part, we first show the channel-suppressing effect and robustness of our CAS, then analyze the effectiveness of our CAS when applied at different layers of DNN. The parameter analysis of CAS can be found in Appendix D. In Appendix E, we show that CAS can also help representation learning and natural training.

**Experimental Settings.** We adversarially train ResNet-18 for 200 epochs on CIFAR-10 using SGD with momentum 0.9, weight decay 2e-4, and initial learning rate 0.1 which is divided by 10 at 75-th and 90-th epoch. We use PGD-10 ($\epsilon = 8/255$ and step size $2/255$) with random start for training. The robustness (accuracy on adversarial examples) is evaluated under attacks: FGSM (Goodfellow et al., 2015), PGD-20 (Madry et al., 2018), and CW$_\infty$ (Carlini & Wagner, 2017) optimized by PGD.

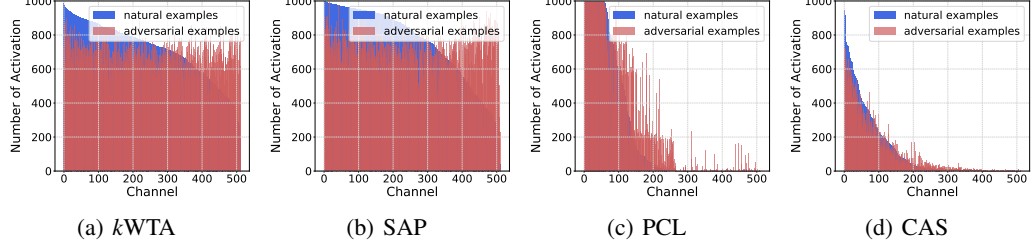

(a) $k$WTA      (b) SAP      (c) PCL      (d) CAS

Figure 4: Comparisons of activation frequency distribution between adversarial and natural examples on different activation or feature oriented defense methods ($k$WTA, SAP, PCL and our CAS).

**Channel Suppressing Effect.** We compare CAS with three activation- or feature-based defense methods: $k$WTA (Xiao et al., 2020), SAP (Dhillon et al., 2018) and PCL (Mustafa et al., 2019). Here, we train $k$WTA with 20% sparse largest values in each activation layer, SAP with typical random pruning and PCL with warm-up training by CE loss and then fine-tuning with the added PCL loss. Figure 4 shows the activation frequencies at the penultimate layer of ResNet-18 trained by different methods. While $k$WTA, SAP and PCL demonstrate a certain level of channel suppressing, their effects are not as significant as our CAS training. $k$WTA and SAP hardly have the channel suppressing effect (*e.g.* channel #350 - #500 in Figure 4(a) and channel #380 - #500 in Figure

Table 1: Robustness (%) of ResNet-18 trained by different defense (*k*WTA, SAP, PCL and our CAS) on CIFAR-10. Avg-PGD-100 denotes 100-step averaged PGD attack (Tramer et al., 2020).

| Defense | Natural | FGSM | PGD-20 | CW$_\infty$ | Avg-PGD-100 | EOT |
|---------|---------|------|--------|-------------|-------------|-----|
| *k*WTA | 76.48 | 59.56 | **50.72** | 46.84 | 16.72 | – |
| SAP | 79.13 | 59.04 | 46.35 | 46.65 | – | 19.98 |
| PCL | **88.15** | 46.47 | 24.68 | 37.50 | – | – |
| CAS | 86.79 | **61.23** | 48.88 | **53.33** | **53.20** | **56.47** |

Table 2: Effectiveness of the channel suppressing operation in CAS module on CIFAR-10 with ResNet-18. CAS is inserted at Block4 of ResNet-18. **Without** suppressing means the CAS module is inserted, however, the channel suppressing operation is not applied during either training or testing. In this case, CAS is just a simple auxiliary classifier.

| Defense | Natural | FGSM | PGD-20 | CW$_\infty$ |
|---------|---------|------|--------|-------------|
| AT | 84.27 | 60.46 | 46.50 | 48.97 |
| AT+**CAS** (**without** suppressing) | 83.42 | 59.81 | 44.20 | 46.27 |
| AT+**CAS** (**with** suppressing) | **86.79** | **61.23** | **48.88** | **53.33** |

4(b)), for the reason that they improve robustness mainly by introducing certain randomness into the activation, and thus can be easily attacked by some adaptive attacks (Tramer et al., 2020; Athalye et al., 2019). PCL still frequently activates many redundant channels (*e.g.* channel #150 - #250 in Figure 4(c)). This is because PCL does not directly enforce channel suppression. Different from these methods, our CAS demonstrates the most effective channel suppression. As a side note, due to the dynamic thresholding, the frequency distributions should be compared within the same model between natural and adversarial examples not across different models. Within the same model, the closer the frequency distribution of the adversarial examples is to that of the natural examples, the better the adversarial robustness (*i.e.* the adversarial accuracy is closer to the natural accuracy).

From this point of view, our CAS can effectively reduce the activation frequency gaps between natural and adversarial examples, producing superior robustness. The natural accuracy and robustness of these methods are reported in Table 1. Due to the randomness introduced in *k*WTA and SAP, they are not robust against average PGD (Avg-PGD) using margin loss (Tramer et al., 2020) or Expectation Over Transformation (EOT) (Athalye et al., 2019) attacks. Our CAS training strategy does not rely on randomness, thus is robust even against Avg-PGD or EOT attacks.

We next verify that explicit Channel Suppressing (CS) is indeed essential to the improved robustness of CAS. Specifically, we remove CS defined in Equation 3 from CAS, then retrain ResNet-18 using adversarial training with the CAS loss defined in Equation 5. Table 2 shows that the robustness can not be improved without explicit channel suppressing.

**CAS at Different Layers.** We insert the CAS module into different blocks of ResNet-18, and show the different robustness improvements in Table 3. Intuitively, deeper layer activation are more correlated to the class prediction, thus should benefit more from our CAS training. Shallow layers, however, may suffer from inaccurate channel importance estimations. As demonstrated in Table 3, it is indeed the case: the largest improvement is obtained when applying CAS at Block4 (*e.g.* after the ReLU output of Block4). The robustness can also be improved when inserting CAS into Block3 or both the Block3 and Block4 ('Block3+4'), though notably less significant than that at Block4.

Table 3: Effectiveness of our CAS module at different blocks of ResNet-18 on CIFAR-10.

| Defense | Block | Natural | FGSM | PGD-20 | CW$_\infty$ |
|---------|-------|---------|------|--------|-------------|
| | Block2 | 71.89 | 49.69 | 40.26 | 46.46 |
| AT+**CAS** | Block3 | 83.05 | 59.20 | 47.84 | 48.19 |
| | Block4 | **86.79** | **61.23** | **48.88** | 53.33 |
| | Block3+4 | 83.77 | 58.32 | 48.27 | **54.62** |

**Robustness of the CAS Module.** Since our CAS module suppresses channel-wise activation according to the label (during training) or the prediction (during testing), it might raise the concerns of

whether the CAS module itself is robust or how the misclassification in CAS module would affect the final results. One observation is that, when the CAS module is inserted to deep layers that are close to the final layer of the network, it can learn to make very similar predictions with the final layer. For an empirical analysis, we test the robustness of the CAS module against different attacks in Table 4. The results indicate that our CAS module is robust by itself, leading to both higher natural accuracy and adversarial robustness. More evaluations can be found in Appendix F.5.

Table 4: Robustness of defense ResNet-18 models trained with (**+CAS**) or without CAS module on CIFAR-10 against different attacks. For **+CAS** models, we only apply the attack on the CAS module using the CAS loss (Equation 4). For baseline defenses, we attack the final layer of the model.

| Defense | Attack Place | Natural | FGSM | PGD-20 | $CW_\infty$ |
|---|---|---|---|---|---|
| AT / **+CAS** | Final / CAS | 84.27/**84.95** | 60.46/**61.40** | 46.50/**47.99** | 48.97/**57.79** |
| TRADES / **+CAS** | Final / CAS | 83.50/**83.84** | 63.68/**64.31** | 52.80/**54.11** | 50.90/**64.01** |
| MART / **+CAS** | Final / CAS | 82.16/**84.89** | 63.91/**65.14** | 52.67/**54.48** | 49.44/**66.92** |

## 5.2 ROBUSTNESS EVALUATION

In this section, we evaluate our CAS on CIFAR-10 (Krizhevsky et al., 2009) and SVHN (Netzer et al., 2011) datasets with ResNet-18 (He et al., 2016). We apply our CAS training strategy to several state-of-the-art adversarial training approaches: 1) AT (Adversarial Training) (Madry et al., 2018), 2) TRADES (Zhang et al., 2019), and 3) MART (Wang et al., 2020c). We follow the default settings as stated in their papers. More results on WideResNet-34-10 (Zagoruyko & Komodakis, 2016) and VGG16 (Simonyan & Zisserman, 2014) can be found in Appendix F.1 and F.2.

**Experimental Settings.** The training settings for CIFAR-10 are the same as Section 5.1. For SVHN, we adversarially train ResNet-18 using SGD with momentum 0.9, weight decay 5e-4, initial learning rate 0.01 which is divided by 10 at 75-th and 90-th epoch, and training attack PGD-10 ( $\epsilon = 8/255$ and step size $1/255$) with random start.

**White-box Robustness.** We evaluate the robustness of all defense models against three types of white-box attacks: FGSM, PGD-20 (step size $\epsilon/10$) and $CW_\infty$ (optimized by PGD). To fairly compare our method with baselines, we use *adaptive white-box attack* for our CAS models, *i.e.* the attacks are performed on the joint loss of CE and CAS. Here, we report the robustness of the models obtained at the last training epoch in Table 5. As shown in Table 5, our CAS can improve both the natural accuracy and the robustness of all baseline methods, resulting in noticeably better robustness. The improvement against $CW_\infty$ attack is more significant than against FGSM or PGD-20 attacks. This is because the prediction margins are enlarged by our CAS training with the channel suppression. As shown in Figure 9 (Appendix E), the deep representations learned by CAS-trained models are more compact within each class, while are more separated across different classes. This makes margin-based attacks like $CW_\infty$ more difficult to success.

Robustness results obtained at the best checkpoint throughout the entire training process and the learning curves are provided in Appendix F.3. Our CAS can also improve the robustness of the best checkpoint model for each baseline defense. Thus the improvement of our CAS is reliable and consistent, and is not caused by the effect of overfitting (Rice et al., 2020). We have also evaluated our CAS training strategy under AutoAttack (Croce & Hein, 2020b) and different attack perturbation budget $\epsilon$ in Appendix F.4 and F.5.

**Black-box Robustness.** We evaluate the black-box robustness of CAS and the baseline methods against both transfer and query-based attacks. For transfer attack, the adversarial examples are generated on CIFAR-10/SVHN test images by applying PGD-20 and $CW_\infty$ attacks on a naturally trained ResNet-50. For query-based attack, we adopt $\mathcal{N}$Attack (Li et al., 2019). Since $\mathcal{N}$Attack requires a lot of queries, we randomly sample 1,000 images from CIFAR-10/SVHN test set and limit the maximum query to 20,000. We test both black-box attacks on the models obtained at the last training epoch. The results are reported in Table 6. For both transfer and query-based attacks, our CAS can improve the robustness of all defense models by a considerable margin. Especially against the $\mathcal{N}$Attack, our CAS training strategy can improve AT, TRADES and MART by $\sim 30\%$ - 40%. One reason why our CAS is particularly more effective against $\mathcal{N}$Attack is that $\mathcal{N}$Attack

Table 5: White-box robustness (accuracy (%) on various white-box attacks) on CIFAR-10 and SVHN, based on the last checkpoint of ResNet-18. '**+CAS**' indicates applying our CAS training strategy to existing defense methods. The best results are **boldfaced**.

| Defense | SVHN | | | | CIFAR-10 | | | |
|---|---|---|---|---|---|---|---|---|
| | Natural | FGSM | PGD-20 | $CW_\infty$ | Natural | FGSM | PGD-20 | $CW_\infty$ |
| AT | 89.62 | 65.09 | 42.55 | 50.96 | 84.27 | 60.46 | 46.50 | 48.97 |
| **AT+CAS** | **90.39** | **67.51** | **51.98** | **53.53** | **86.79** | **61.23** | **48.88** | **53.33** |
| TRADES | 91.16 | 69.85 | 50.90 | 50.85 | 83.50 | 63.68 | 52.80 | 50.90 |
| **TRADES+CAS** | **91.69** | **70.97** | **55.26** | **60.10** | **85.83** | **65.21** | **55.99** | **67.17** |
| MART | 91.16 | 67.31 | 48.72 | 50.52 | 82.16 | **63.91** | 52.67 | 49.44 |
| **MART+CAS** | **93.05** | **70.30** | **51.57** | **53.38** | **86.95** | 63.64 | **54.37** | **63.16** |

utilizes a similar margin objective function as $CW_\infty$, which can be effectively blocked by channel suppressing.

Table 6: Black-box robustness (accuracy (%) on various black-box attacks) of ResNet-18 on SVHN and CIFAR-10. '**+CAS**' indicates applying our CAS training strategy to existing defense methods. The best results are **boldfaced**.

| Defense | SVHN | | | CIFAR-10 | | |
|---|---|---|---|---|---|---|
| | PGD-20 | $CW_\infty$ | $\mathcal{N}$Attack | PGD-20 | $CW_\infty$ | $\mathcal{N}$Attack |
| AT | 64.42 | 70.60 | 40.36 | 79.13 | 79.87 | 45.05 |
| **AT+CAS** | **65.50** | **72.35** | **75.72** | **85.80** | **86.54** | **83.32** |
| TRADES | 67.81 | 74.43 | 44.16 | 78.18 | 78.95 | 49.25 |
| **TRADES+CAS** | **68.48** | **75.66** | **81.82** | **84.77** | **85.54** | **79.42** |
| MART | 66.85 | 73.90 | 41.66 | 76.93 | 77.49 | 49.05 |
| **MART+CAS** | **68.45** | **75.45** | **80.12** | **85.68** | **86.59** | **81.93** |

## 6 CONCLUSION

In this paper, we investigated intermediate activation of deep neural networks (DNNs) from a novel channel-wise perspective, in the context of adversarial robustness and adversarial training. We highlight two new characteristics of the channels of adversarial activation: 1) higher magnitude, and 2) more uniform activation frequency. We find that standard adversarial training improves robustness by addressing the first issue of the higher magnitude, however, it fails to address the second issue of the more uniform activation frequency. To overcome this, we proposed the Channel-wise Activation Suppressing (CAS), which dynamically learns the channel importance and leverages the learned channel importance to suppress the channel activation in the training phase. When combined with adversarial training, we show that, CAS can train DNNs that inherently suppress redundant channels from being activated by adversarial examples. Our CAS is a simple but generic training strategy that can be easily plugged into different defense methods to further improve their robustness, and can be readily applied to robustify the intermediate layers of DNNs.

## ACKNOWLEDGEMENT

Yisen Wang is partially supported by the National Natural Science Foundation of China under Grant 62006153, and CCF-Baidu Open Fund (OF2020002). Shu-Tao Xia is partially supported by the National Key Research and Development Program of China under Grant 2018YFB1800204, the National Natural Science Foundation of China under Grant 61771273, the R&D Program of Shenzhen under Grant JCYJ20180508152204044.

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

## A    ALGORITHM OF CAS TRAINING

---

**Algorithm 1** Robust Training with CAS.

---

**Input:**  Training data $\{\boldsymbol{x}_i, y_i\}_{i=1,2,\dots,n}$, DNN $\mathcal{F}(\theta)$, CAS modules with parameters $M$, maximum training epochs $T$
**Output:**  Robust Network $\mathcal{F}$
 1: **for** $t$ in $[1, 2, \cdots, T]$ **do**
 2:     **for** minibatch $\{\boldsymbol{x}_1, \cdots, \boldsymbol{x}_b\}$ **do**
 3:         Generate adversarial examples using PGD attack on Equation 5
 4:         Compute the CAS loss in Equation 4 using the channel-wise activation $\hat{\boldsymbol{f}}^l$
 5:         Reweight the original features $\tilde{\boldsymbol{f}}^l = \boldsymbol{f}^l \otimes M_y^l$ using the parameters $M_y^l$ in CAS
 6:         Forward with the adjusted $\tilde{\boldsymbol{f}}^l$ and compute the CE loss at the output layer
 7:     **end for**
 8:     Optimize all parameters $(\theta, M)$ by Equation 5 using gradient descent
 9: **end for**

---

## B    A SUMMARY OF THE ADVERSARIAL LOSS FUNCTIONS USED WITH CAS

When combined with our CAS, the training loss is a combination of the original adversarial loss and our CAS loss. Table 7 defines the exact loss functions used for AT (Madry et al., 2018), TRADES (Zhang et al., 2019), MART (Wang et al., 2020c) and their CAS enhanced versions. Here, we assume the CAS module is attached to $S$ number of layers of the network. $\hat{\boldsymbol{p}}^s$ and $M^s$ denote the prediction score and weights of the auxiliary classifier in the $s$-th CAS module, respectively.

Table 7: A summary of the loss functions used for standard adversarial training (AT), TRADES, MART, and their corresponding versions with our CAS ('**+CAS**').

| Defense | Loss Function |
|---|---|
| AT | $\mathrm{CE}(\boldsymbol{p}(\boldsymbol{x}', \theta), y)$ |
| **+CAS** | $+\frac{\beta}{S} \sum_{s=1}^{S} \mathrm{CE}(\hat{\boldsymbol{p}}^s(\boldsymbol{x}', \theta, M^s), y)$ |
| TRADES | $\mathrm{CE}(\boldsymbol{p}(\boldsymbol{x}, \theta), y) + \lambda \cdot \mathrm{KL}(\boldsymbol{p}(\boldsymbol{x}, \theta) || \boldsymbol{p}(\boldsymbol{x}', \theta))$ |
| **+CAS** | $+\frac{\beta}{S} \sum_{s=1}^{S} \mathrm{CE}(\hat{\boldsymbol{p}}^s(\boldsymbol{x}, \theta, M^s), y) + \beta \cdot (\frac{\lambda}{S} \cdot \sum_{s=1}^{S} \mathrm{KL}(\hat{\boldsymbol{p}}^s(\boldsymbol{x}, \theta, M^s) || \hat{\boldsymbol{p}}^s(\boldsymbol{x}', \theta, M^s)))$ |
| MART | $\mathrm{BCE}(\boldsymbol{p}(\boldsymbol{x}', \theta), y) + \lambda \cdot \mathrm{KL}(\boldsymbol{p}(\boldsymbol{x}, \theta) || \boldsymbol{p}(\boldsymbol{x}', y)) \cdot (1 - \boldsymbol{p}_y(\boldsymbol{x}, \theta))$ |
| **+CAS** | $+\frac{\beta}{S} \sum_{s=1}^{S} \mathrm{BCE}(\hat{\boldsymbol{p}}^s(\boldsymbol{x}', \theta, M^s), y) + \beta \cdot (\frac{\lambda}{S} \cdot \sum_{s=1}^{S} \mathrm{KL}(\hat{\boldsymbol{p}}^s(\boldsymbol{x}, \theta, M^s) || \hat{\boldsymbol{p}}^s(\boldsymbol{x}', \theta, M^s)) \cdot (1 - \hat{\boldsymbol{p}}_y^s(\boldsymbol{x}, \theta, M^s))$ |

## C    CHANNEL-WISE ACTIVATION SUPPRESSING ON MORE DATASETS AND DEFENSE MODELS

Here, we visualize the channel suppressing effect of our CAS training strategy on more defense models: TRADES (Zhang et al., 2019) and MART (Wang et al., 2020c). We train ResNet-18 (He et al., 2016) on CIFAR-10 (Krizhevsky et al., 2009). The CAS modules are attached to the final block of ResNet-18.

Figure 5 illustrates the channel activation frequencies of the original defense models and their improved versions with our CAS. As can be observed, our CAS can suppress the redundant channels consistently in both defense models. This restricts the channel-wise activation to be more class-correlated. More importantly, the channel activation are suppressed to a similar frequency distribution between the natural and the adversarial examples.

We further compare the channel-wise activation of natural versus adversarial examples on SVHN and ImageNet in Figure 6 and Figure 7. For SVHN, we choose examples of class 0 and construct the adversarial examples by PGD-20 with $\epsilon = 8/255$. We count the channel activation frequency of naturally trained, adversarially trained and CAS trained models. As shown in Figure 6, neurons #200 - #500 in Figure 6(a) and neurons #250 - #400 in Figure 6(b), are frequently activated by adversarial examples while in Figure 6(c), our CAS can effectively suppress these redundant activation.

For ImageNet, as it is extremely time consuming to perform adversarial training, we adopt the standard pre-trained ResNet-152 from torchvision.models * and an adversarially trained ResNet-152 from the Github repository[†]. We select 5,900 dog images from the validation set of ImageNet (class #151-#269) and generate the adversarial examples using PGD-30 ($\epsilon$=16/255, $\alpha$=1/255). We count the channel activation at the penultimate layer of ResNet-152. It can be observed in Figure 7(a) that adversarial examples activate more uniformly than natural examples. While in Figure 7(b), adversarially trained models can align the activation distribution of the natural examples with that of the adversarial examples to some extent, however, the activation frequencies of the adversarial examples are still higher than that of the natural examples. This indicates that there are still redundant channels and non-robust features in this adversarially trained ResNet-152 model.

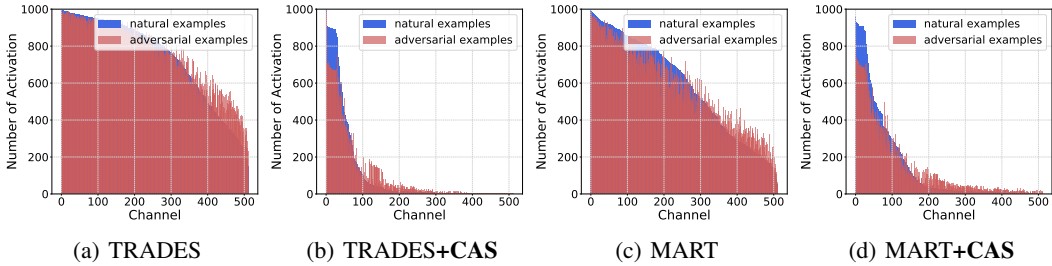

Figure 5: The distributions of channel activation frequency of both natural and adversarial examples in different defense models (*e.g.* TRADES and MART). The frequency distribution gap between natural and adversarial examples is effectively narrowed down by our CAS training, and the redundant channels (channel #200 - #512) are significantly suppressed by CAS.

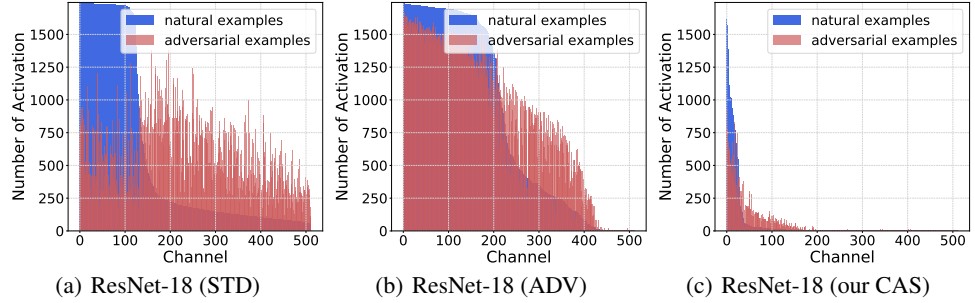

Figure 6: The distributions of channel activation frequency of both natural and PGD-20 adversarial examples for ResNet-18 on SVHN.

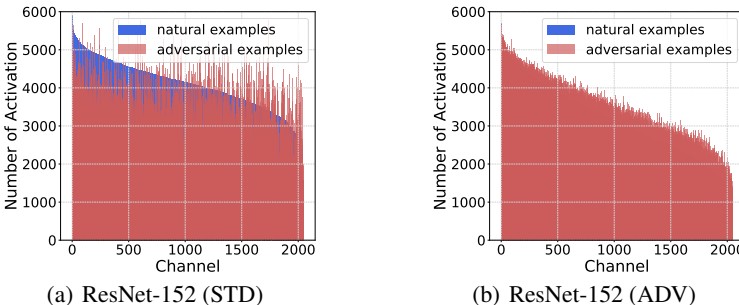

Figure 7: The distributions of channel activation frequency of both natural and PGD-30 adversarial examples for ResNet-152 on ImageNet.

*https://pytorch.org/docs/stable/torchvision/models.html

[†]https://github.com/facebookresearch/ImageNet-Adversarial-Training

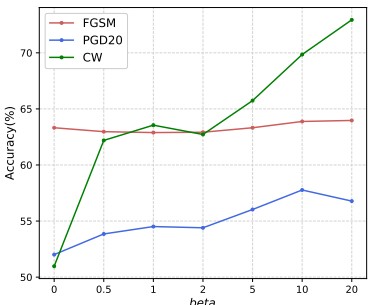

Figure 8: Robustness of AT+CAS against white-box attacks FGSM, PGD-20 and $CW_\infty$ under different $\beta$. As $\beta$ increases, the robustness is also improved, especially against the $CW_\infty$ attack.

## D  SENSITIVITY OF CAS TO PARAMETER $\beta$

As mentioned in Section 4, the parameter $\beta$ on the CAS loss controls the suppressing strength. To test the sensitivity of CAS training under different $\beta$, we insert a CAS module to Block4 of ResNet-18, and train the network on CIFAR-10 using AT+CAS under $\beta \in [0, 0.5, 1, 2, 5, 10, 20]$. Note that $\beta = 0$ indicates the standard adversarial training (AT). We test the robustness of the models in a white-box setting against FGSM, PGD-20 and $CW_\infty$. As shown in Figure 8, the models trained with larger $\beta$ are generally more robust, especially against the $CW_\infty$ attack. This is because larger $\beta$ increases the strength of channel suppressing, leading to larger inter-class margins. As we further show in Figure 9, the representations learned with CAS are more separated between different classes, and are more compact within the same class. This tends to increase the difficulty of margin-based attacks like white-box $CW_\infty$ attack and black-box $\mathcal{N}$ attack.

## E  CAS IMPROVES REPRESENTATION LEARNING

The representations learned by natural or adversarial training with or without our CAS strategy are illustrated in Figure 9. The t-SNE (Maaten & Hinton, 2008) 2D embeddings are computed on deep features extracted at the penultimate layer of ResNet-18 on CIFAR-10. As can be observed, our CAS training improves representation learning for both natural training and adversarial training. This is largely attributed to the strong channel suppressing capability of our CAS training. Channel suppressing helps learn high-quality representations with high inter-class separation and intra-class compactness. Interestingly, our CAS training can even improve the performance of natural training from 92.75% to 94.56%. This implies that our CAS is a generic training strategy that can benefit both model training and representation learning. Although CAS is not a direct regularization technique, it can achieve a similar representation regularization effect as existing representation enhancing techniques like the center loss (Wen et al., 2016).

## F  MORE EXPERIMENTAL RESULTS

### F.1  WIDERESNET RESULTS ON CIFAR-10

The white-box robustness of WideResNet-34-10 (Zagoruyko & Komodakis, 2016) models trained using AT, AT+CAS, TRADES, TRADES+CAS, MART and MART+CAS are reported in Table 8. We attach the CAS module to the last two convolution layers of the network, and set $\beta = 2$. The training settings are the same as used for ResNet-18, except that, here we use weight decay 5e-4. The 'best' and 'last' results indicate the best and last checkpoints, respectively. For our CAS, we generate the attacks using the same *adaptive white-box attacks* as used in Section 5.2. The robustness of AT, TRADES and MART can be consistently improved by our CAS training. The most improvements are achieved on AT. These results confirm that our CAS training can lead to consistent improvements to different adversarial training methods on more complex models.

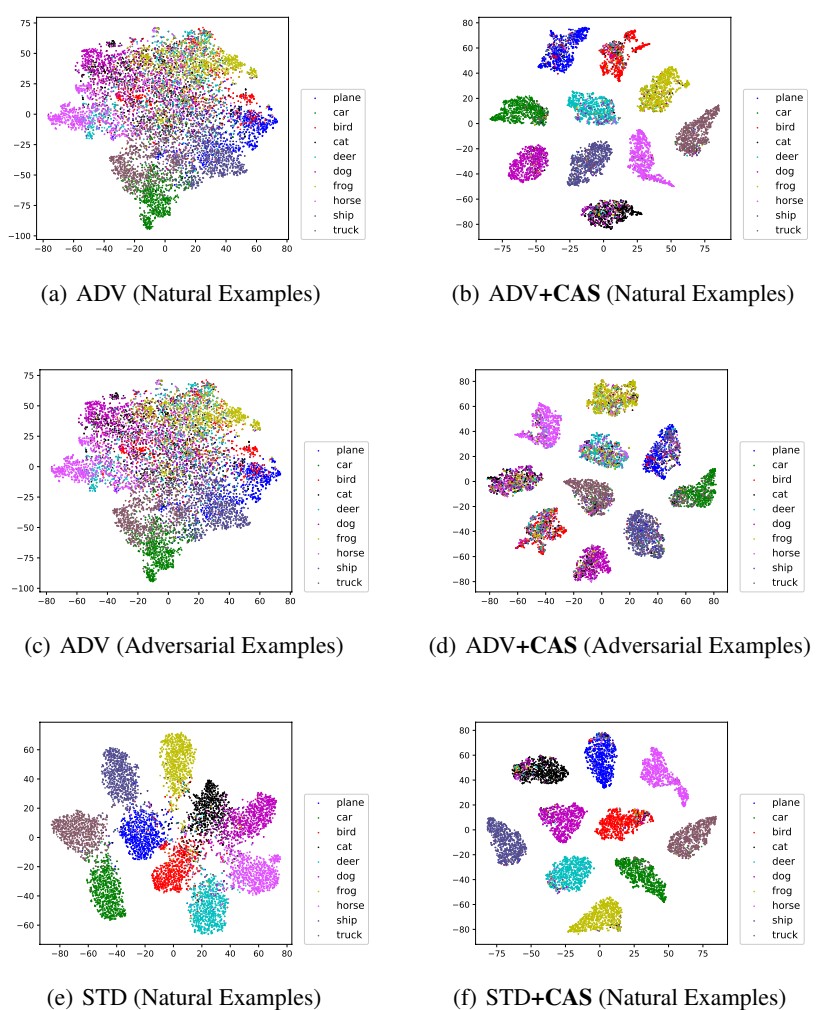

(a) ADV (Natural Examples)

(b) ADV**+CAS** (Natural Examples)

(c) ADV (Adversarial Examples)

(d) ADV**+CAS** (Adversarial Examples)

(e) STD (Natural Examples)

(f) STD**+CAS** (Natural Examples)

Figure 9: The t-SNE 2D embeddings of deep features extracted at the penultimate layer of ResNet-18 models trained using natural ('STD') or adversarial training ('ADV') on CIFAR-10. The embeddings are shown separately for natural versus adversarial examples. Our CAS training can help improve inter-class separation and intra-class compactness for both types of training.

Table 8: White-box robustness (accuracy (%) on various white-box attacks) of WideResNet-34-10 on CIFAR-10 dataset. '**+CAS**' indicates applying our CAS training to existing defense methods. The best results are **boldfaced**.

| Defense | CIFAR-10 | | | | | | | |
|---|---|---|---|---|---|---|---|---|
| | Natural | | FGSM | | PGD-20 | | CW$_\infty$ | |
| | Best | Last | Best | Last | Best | Last | Best | Last |
| AT | 84.16 | 84.23 | 66.62 | 61.27 | 53.75 | 49.53 | 50.52 | 47.68 |
| AT**+CAS** | **88.25** | **89.06** | **67.14** | **66.98** | **56.28** | **53.22** | **58.54** | **54.77** |
| TRADES | 86.23 | 86.41 | 66.41 | 65.63 | 54.42 | 52.64 | 53.45 | 52.43 |
| TRADES**+CAS** | **87.07** | **87.15** | **66.92** | **66.15** | **55.43** | **53.15** | **61.46** | **57.07** |
| MART | 84.09 | 85.69 | 67.24 | 66.41 | 57.56 | 54.49 | 54.42 | 52.55 |
| MART**+CAS** | **87.87** | **89.20** | **68.09** | **67.96** | **58.24** | **54.95** | **61.48** | **57.11** |

### F.2 VGG16 RESULTS ON CIFAR-10

Similar to the above WideResNet experiments, here we report the results of VGG16 (Simonyan & Zisserman, 2014) on CIFAR-10 in Table 9. For VGG16, we attach the CAS module to its last three convolution layers, and set $\beta = 3$. We can see that our CAS training can enhance the robustness of 'best' models by a remarkable margin of 7%-10%. Note that the accuracy of natural training is also improved by a considerable margin. Compared with complex models like WideResNet, our CAS training is even more beneficial for small capacity models (*e.g.* VGG16).

Table 9: White-box robustness (accuracy (%) on various white-box attacks) of VGG16 on CIFAR-10. '**+CAS**' indicates applying our CAS training strategy to existing defense methods. The best results are **boldfaced**.

| Defense | CIFAR-10 | | | | | | | |
| | Natural | | FGSM | | PGD-20 | | CW$_\infty$ | |
| | Best | Last | Best | Last | Best | Last | Best | Last |
|---|---|---|---|---|---|---|---|---|
| AT | 70.32 | 70.38 | 51.87 | 51.85 | 42.23 | 42.01 | 43.63 | 43.81 |
| AT**+CAS** | **82.25** | **82.61** | **59.77** | **57.56** | **49.22** | **42.73** | **53.03** | **49.68** |
| TRADES | 74.98 | 76.67 | 52.10 | 52.99 | 41.53 | 41.13 | 45.50 | 45.25 |
| TRADES**+CAS** | **83.45** | **83.25** | **61.36** | **61.51** | **49.86** | **49.56** | **52.60** | **52.47** |
| MART | 71.70 | 72.20 | 54.72 | 54.90 | 46.55 | 46.53 | 44.52 | 44.72 |
| MART**+CAS** | **81.53** | **83.28** | **61.89** | **61.70** | **52.01** | **49.13** | **50.97** | **49.10** |

### F.3 ROBUSTNESS RESULTS AT THE BEST CHECKPOINT

We report the white-box robustness of the 'best' (*i.e.* the best checkpoint) models of ResNet-18 on SVHN and CIFAR-10 in Table 10, as a supplementary to the 'last' (*i.e.* the last checkpoint) results in Table 5. Again, our CAS can also improve the robustness of the 'best' checkpoint models consistently. This proves that our CAS can improve both the robustness and the natural accuracy throughout the entire training process.

Table 10: White-box robustness (accuracy (%) on various white-box attacks) of ResNet-18 on CIFAR-10 and SVHN on the best checkpoint. '**+CAS**' indicates applying our CAS training strategy to existing defense methods. The best results are **boldfaced**.

| Defense | SVHN | | | | CIFAR-10 | | | |
| | Natural | FGSM | PGD-20 | CW$_\infty$ | Natural | FGSM | PGD-20 | CW$_\infty$ |
|---|---|---|---|---|---|---|---|---|
| AT | 91.41 | 69.52 | 53.07 | 50.38 | 84.20 | **63.32** | 52.01 | 50.97 |
| AT**+CAS** | **92.69** | **71.83** | **54.78** | **54.13** | **86.72** | 62.92 | **54.40** | **62.72** |
| TRADES | 90.40 | 71.22 | 57.71 | 54.49 | 82.68 | 63.15 | 53.05 | 50.46 |
| TRADES**+CAS** | **91.41** | **71.39** | **59.32** | **66.94** | **84.75** | **63.94** | **57.20** | **67.77** |
| MART | 87.37 | 68.58 | 57.72 | 51.48 | 78.90 | 63.29 | 54.92 | 50.31 |
| MART**+CAS** | **91.04** | **72.32** | **60.29** | **57.05** | **85.99** | **64.23** | **58.21** | **69.91** |

### F.4 ROBUSTNESS AGAINST AUTOATTACK

We report the white-box AutoAttack (Croce & Hein, 2020b) evaluation results of ResNet-18 on CIFAR-10 in Table 11. AutoAttack is an ensemble of two proposed Auto-PGD attacks and the other two complementary attacks (Croce & Hein, 2020a). AutoAttack has been shown can produce more accurate robustness evaluations on a wide range of adversarial training defenses. As shown in Table 11, although less significant than against regular attacks like PGD and CW, CAS with proper training can still improve the robustness of AT, TRADES and MART by a noticeable margin. This confirms that the improvements brought by our CAS training is 'real' and substantial, rather than obfuscated gradients nor improper evaluation.

Table 11: White-box robustness (accuracy (%) against AutoAttack) of ResNet-18 on CIFAR-10. '**+CAS**' indicates applying our CAS training strategy to existing defense methods. The best results are **boldfaced**.

| Defense | AutoAttack | |
| --- | --- | --- |
| | Best | Last |
| AT | 46.58 | 41.90 |
| AT**+CAS** | **47.40** | **44.74** |
| TRADES | 48.28 | 47.46 |
| TRADES**+CAS** | **48.40** | **48.38** |
| MART | 47.06 | 45.45 |
| MART**+CAS** | **48.45** | **46.39** |

### F.5    MORE ROBUSTNESS EVALUATIONS OF CAS

The robustness of the CAS module is further evaluated in this section. By producing large margins, the CAS module and the channel suppression operation can make more boundary adversarial examples be correctly classified than that in the baseline models. However, this may also increase the risk of causing imbalanced margin and imbalanced gradients. To test this, we evaluate CAS against a recent Margin Decomposition (MD) attack (Jiang et al., 2020). We use the default setting of MD attack as stated in the original paper. The results are reported in Table 12, where it shows our CAS is robust against the MD attack. We also test our CAS against PGD-20 with step size $\epsilon/10$ and various attack strengths. This result is presented in Table 13. The robustness of CAS decreases with the increase of $\epsilon$, which proves that the robustness of CAS is not a result of obfuscated gradients.

Table 12: Robustness of CAS against Margin Decomposition (MD) attack. This experiment was conducted with ResNet-18 on CIFAR-10.

| Defense | Adversarial Loss Type | | |
| --- | --- | --- | --- |
| | MD (CE) | MD (CAS) | MD (CE+CAS) |
| AT | 43.35 | – | – |
| AT**+CAS** | – | **46.27** | **49.56** |
| TRADES | 48.47 | – | – |
| TRADES**+CAS** | – | **53.51** | **55.53** |
| MART | 46.99 | – | – |
| MART**+CAS** | – | **49.89** | **55.75** |

Table 13: Robustness (%) of CAS with ResNet-18 on CIFAR-10 against PGD-20 under different $\epsilon$.

| Attack strength $\epsilon$ | 2/255 | 4/255 | 8/255 | 16/255 | 32/255 |
| --- | --- | --- | --- | --- | --- |
| AT**+CAS** | 79.15 | 70.26 | 48.88 | 16.64 | 2.48 |
| TRADES**+CAS** | 80.09 | 72.69 | 55.99 | 27.94 | 5.10 |
| MART**+CAS** | 81.01 | 73.56 | 54.37 | 20.49 | 2.19 |

### F.6    CHANNEL ACTIVATION FREQUENCY UNDER DIFFERENT THRESHOLDS

In Figure 2, we have shown the activation frequency of standard training (STD), adversarial training (AT) and CAS-based adversarial training (AT+CAS) on a ResNet-18 model with the threshold of 1%. That is, a channel is determined as activated if its activation count is larger that 1% of the maximum activation count over all 512 channels. In Figure 10, we show the distribution of activation frequency under different thresholds (*e.g.* 0.5%, 1%, 5%). Larger threshold will truncate more low-count activation. The plots show that different thresholds will introduce some noise to the distributions but the general patterns do not change.

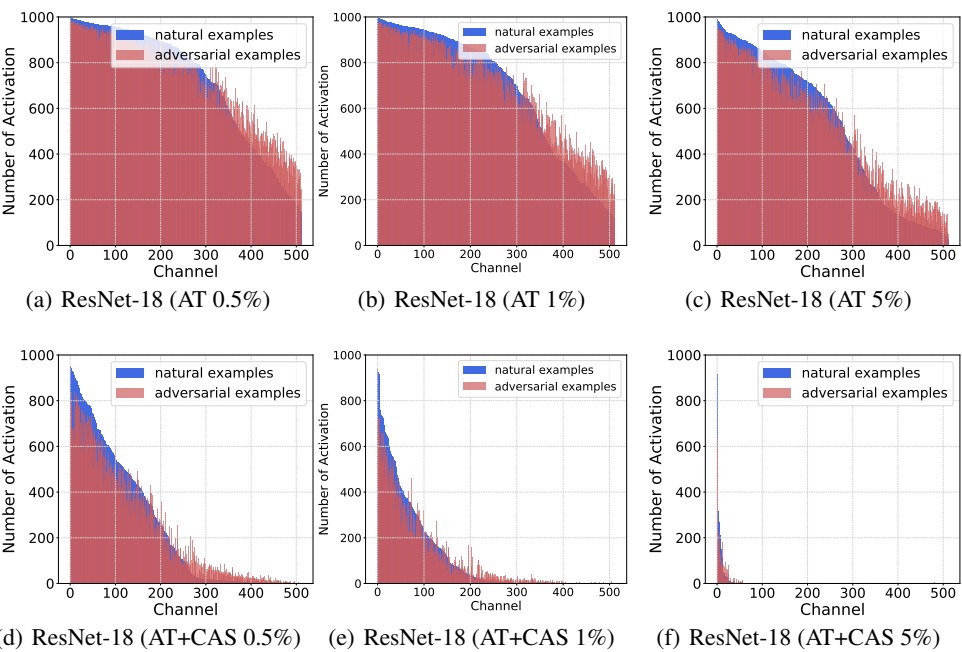

Figure 10: The activation frequency (y-axis) of channel-wise activation at the penultimate layer of adversarial training (AT) and our CAS-based adversarial training (AT+CAS) for ResNet-18 on CIFAR-10. The activation frequencies are visualized with respect to different thresholds (*e.g.* 0.5%, 1% and 5%).

