# OpenReview forum: "Improving Adversarial Robustness via Channel-wise Activation Suppressing"
_ICLR.cc/2021/Conference — ICLR 2021 Spotlight_

### Official Review · AnonReviewer4 · 2020-10-27
**This paper proposes to use channel suppressing to enhance adversarial training.**

**Rating:** 7
**Confidence:** 5

**Review:**

##########################################################################
Summary:

This paper uncovers interesting phenomenons of adversarial training, i.e., more uniformly distributed adversarial data activations than those of natural data. To force the behaviors (in this paper, channels activations) of adversarial data to be similar to those of natural data, the authors explicitly suppress the redundant channels by reweighing the channel activations.

##########################################################################
Reason for score.

Overall, I vote for accepting. I like the uncovered phenomenons of larger and more uniformly distributed activations of adversarial data than those of natural data.
Technically, this paper proposed effective training strategies (i.e., channel-wise activation suppressing (CSA)) to enhance adversarial training.

##########################################################################
Pros:

1 This paper provides the understanding of adversarial training from the channel activation perspective, showing that adversarial training can reduce the magnitude of the activation of the adversarial data, but fail to break the uniform activations by the adversarial data.

2 Figure 2 shows the efficacy of the proposed CSA methods for breaking the adversarial data's uniform activations. Compared with standard adversarial training, CSA can further suppress the redundant channel activations.

3 The experiment evaluations are comprehensive, showing CSA strategies' efficacy across various adversarial training methods, network structures, and attack methods.

##########################################################################
Cons:

1 What is the side effect of redundant channel activations? Specifically, what is the side effect of uniform activations of the adversarial data? Would you mind explaining more?

2 Although CSA successfully suppresses the redundant channels of the adversarial data, CSA also seems to suppress the activations of natural data? Is this the reason for the improvement on natural accuracy?

---

> ### Author Response · Authors · 2020-11-20
> **Response to AnonReviewer4**
>
> Thanks for your valuable comments. Please find our responses below.
>
> ---
> **Q1:** What is the side effect of redundant channel activations? Specifically, what is the side effect of uniform activations of the adversarial data?
>
> **A1:** As high-level features are highly correlated to the predicted class, the redundant features can easily cause mispredictions to the wrong classes, that is, poor robustness. Our experiments indicate that this negative effect is more associated with the channels.
>
> ---
> **Q2:** Although CAS successfully suppresses the redundant channels of the adversarial data, CAS also seems to suppress the activations of natural data? Is this the reason for the improvement on natural accuracy?
>
> **A2:** Yes, we find that suppressing redundant channels can lead to more separable representations between different classes, and more compact representations within the same class, as we show in Figure 9 (e) and (f). Representations that are of high intra-class compactness and high inter-class separation have been shown can improve the natural accuracy [1].
>
> [1] Wen, Yandong, et al. "A discriminative feature learning approach for deep face recognition." In ECCV, 2016.
>
> ---

---

> > ### Comment · AnonReviewer4 · 2020-11-23
> > **Thanks for the response.**
> >
> > I went through other reviewers' comments and the replies.
> > The authors have clarified my concerns, and I support acceptance.

---

### Official Review · AnonReviewer2 · 2020-10-27
**Easy to understand and interesting while requires more explanations**

**Rating:** 7
**Confidence:** 4

**Review:**

This paper investigates the adversarial robustness from the activation perspective. Specifically, the authors analyzed the difference in the magnitude and distribution of activation between adversarial examples and clean examples: the activation magnitudes of adversarial examples are higher and the activation channels are more uniform by adversarial examples. Based on the above interesting findings, the authors claim that different channels of intermediate layers contribute differently to the class prediction and propose a Channel-wise Activation Suppressing (CAS) method to suppress redundant activations, which can improve the DNN robustness.

Some highlights in this paper:
+ The CAS strategy is simple and can be easily applied to existing models. Combining CAS with the existing adversarial training methods leads to better DNN robustness.
+ The experiments are well-conducted and convincing. The authors not only provided ablation experiments to verify the effectiveness of CAS, but also provided both the performance of the last epoch and the performance of early stop, which confirmed that CAS can improve the DNN robustness.
+ The paper is well-written and the idea is easy to follow.

However, there are some downsides. I’d like more details about:
- Adversarial training inhibits the magnitude of activation, what is the connection between this and network robustness?
- The closer the activation distribution of the adversarial example is to that of the clean example, the better the robustness of the network. It would be good to provide more discussions and explanations here.

Overall the paper is easy to understand and interesting.

---

> ### Author Response · Authors · 2020-11-20
> **Response to AnonReviewer2**
>
> Thanks for your valuable comments. Please find our responses below.
>
> ---
> **Q1:** Adversarial training inhibits the magnitude of activation, what is the connection between this and network robustness?
>
> **A1:** Activation magnitude is a result of three factors: the norm of the weights, the norm of the input from the previous layer, and the angle between the two vectors. Inhibiting the activation magnitude can reduce the amplification effect of the adversarial perturbation on the current (and the next) layer activation, leading to small change at the output layer, i.e., improved robustness. Note that the activation magnitude of adversarial examples is only inhibited (by adversarial training) to be the same as that of the natural examples.
>
> ---
> **Q2:** More discussions on that the closer the activation distribution of the adversarial example is to that of the clean example, the better the robustness of the network.
>
> **A2:** Channels are correlated to classes. Adversarial examples tend to activate those redundant channels that are correlated to the **wrong** classes. This can easily lead to misclassification and poor robustness. When the activation distribution of the adversarial example is the same as that of the natural example, similar channels will be activated, producing similar class prediction, i.e., good robustness.
>
> ---

---

### Official Review · AnonReviewer1 · 2020-10-27
**This is a novel research paper**

**Rating:** 8
**Confidence:** 4

**Review:**

The authors studied the behavior of adversarial examples from the channel view of activations, which is very novel. They focused on the magnitude and frequency of activations and found that state-of-the-art adversarial defense (adversarial training) only addressed the magnitude issue but the frequency distribution issue remains. This provided a novel perspective for us to understand why state-of-the-art adversarial training method works to a certain extent but not so good. Then, the authors proposed a Channel-wise Activation Suppressing (CAS) to address the frequency distribution to further improve the adversarial robustness. CAS is generic, effective, and can be easily incorporated into many existing defense methods.

Pros:
1. The authors studied adversarial examples from a new perspective of channels in activations. Previous works focusing on activations usually assumed that each channel is of equal importance, while the authors focused on the relationship between channels. From two aspects of activation magnitude and frequency, the authors found two novel characteristics of adversarial examples: adversarial examples have higher activation magnitude and more uniformly activated channels compared to natural examples. The findings were convincingly evaluated on different neural network architectures and different training methods. This hints at a very interesting phenomenon.

2. The proposed method is generic. The authors found that the activated channels are still uniform under adversarial training, that is, some redundant and low contributing channels are still activated. To suppress the redundantly activated channels, the authors proposed Channel-wise Activation Suppressing (CAS) training strategy. It dynamically learns and incorporates the channel importance (to the class prediction) into the training process. The motivation is very clear and the method is easy to follow. More importantly, CAS can be widely applied to strengthen existing adversarial training approaches since it suppresses those less important channels.

3. Lots of experiments are provided to understand and evaluate the proposed methods. The experiments covered lots of aspects, including channel suppressing effect of CAS, representation learning, ablation studies, and extensive robustness evaluation on white-box and black-box attacks. The authors also tested the adaptive attacks, strongest auto-attack, and the optimization-based black-box attack, which definitely convinced me of the effectiveness of the proposed method.

Overall, the paper hints at an interesting phenomenon and inspires an in-depth understanding of adversarial training. The proposed method is elegant and generic. The empirical evidence is solid and extensive.

Cons:
1. How does the activation threshold effect Figure 2?
2. In the testing phase, the predicted class of the auxiliary classifier is used for the channel importance. Is it vulnerable to attacks? if the predicted label is incorrect, how it will affect the final performance?
3. How well the auxiliary classifier works. With the limited information from the output of GAP, it is likely that the classifier performs poorly, and thus results in bad channel importance weighting.
4. CAS could both improve natural acc and adversarial robustness, why CAS could achieve this both? and how is the overhead of CAS?

---

> ### Author Response · Authors · 2020-11-20
> **Response to AnonReviewer1**
>
> Thank you very much for the valuable comments. Please find below our responses to your questions.
>
> ---
> **Q1:** How does the activation threshold affect Figure 2?
>
> **A1:** The thresholding is applied for better visualization. A larger threshold will suppress more low-frequency channels, resulting in a sharper distribution. But the general patterns do not change. We have added the plots under different thresholds in Figure 10, Appendix F.6.
>
> ---
> **Q2:** Is the auxiliary classifier vulnerable to attacks? If the predicted label is incorrect, how it will affect the final performance?
>
> **A2:**  The CAS module itself is robust to attacks. We have added this evaluation in Table 4 and the analysis in Section 5.1 “Robustness of the CAS Module”. The robustness decrease is within 1% compared to PGD-20 on both loss terms. We have also added an evaluation against an adaptive Margin Decomposition (MD) attack in Table 12 and Appendix F.5.
>
> ---
> **Q3:** How well the auxiliary classifier works?
>
> **A3:** We have conducted an ablation study on CAS inserted at different residual blocks of ResNet18 in Table 3. The results show that CAS works the best at the last block (i.e. Block 4), where the auxiliary classifier is as good as the final classification layer. The auxiliary classifier is indeed less accurate at the shallow blocks (e.g. Block 2), where it can cause inaccurate suppression and degraded robustness.
>
> ---
> **Q4:** CAS could both improve natural acc and adversarial robustness, why CAS could achieve this both? and how is the overhead of CAS?
>
> **A4:** The accuracy-robustness trade-off refers to, for the same model, the increase of one metric along with the decrease of the other metric. This is also the case for CAS: natural accuracy drops from 94.56% (Appendix E) to 90.39% (Table 4) while robustness increases from 0% to 48.88% (Table 4).  The overhead is 5% extra training time as compared to standard adversarial training.
>
> ---

---

### Official Review · AnonReviewer3 · 2020-10-28
**Good paper with some technical details to confirm**

**Rating:** 7
**Confidence:** 4

**Review:**

This paper studies the use of channel suppression in improving robustness to adversarial examples. The authors make a convincing illustration in section 3 on how adversarial examples tend to activate more channels compared to natural examples, and adversarial training is not effective in reducing them. This provides a convincing motivation to their design of the Channel-wise Activation Suppression (CAS) module. Their CAS module is also effective in improving adversarial robustness when used in conjunction with different adversarial defense methods, including adversarial training, TRADES, and MART.

I think this paper is of high quality, but I do have several questions on the details:
1. In section 4.1 there is a difference in how the mask M is produced in training and test phase. How important is it to have the correct y available for the mask, as oppose to \hat{y} from the channel predictions? For example it might be difficult to predict the target class from the low-level features (e.g. block 2 channel features), leading to inaccurate \hat{y} for channel suppression. Could this be a reason for lower performance of inserting CAS into block 2 in Table 3?
2. Just to confirm, are both losses (CE and CAS) in Eq 5 taken into account in the generation of adversarial perturbations with FGSM and PGD?
3. In Table 2, what does it mean to have CAS without channel suppression? Is it effectively just a CNN with predictions made from features in different layers?
4. Do the authors have any intuitions on why having CAS module alone on Block 4 is better than having it on both Block 3 and 4 in general?

I am leaning towards acceptance of this paper if the authors can address the above questions sufficiently.

After author response: the authors have sufficiently addressed my questions and also the other reviewers' questions. I am keeping my score of acceptance.

---

> ### Author Response · Authors · 2020-11-20
> **Response to AnonReviewer3**
>
> Thank you very much for the thoughtful comments. Please find our responses below.
>
> ---
> **Q1:** How important is it to have the correct $y$ available for the mask, as opposed to $\hat{y}$ from the channel predictions?
>
> **A1:** We find that an accurate prediction of $y$ is crucial. The class predictions from low-level features are less accurate, and it is indeed the reason why inserting CAS at block 2 has a lower performance. The correlation of the shallow layer channels to the class is low, leading to less accurate channel importance estimation and suppression.
>
> ---
> **Q2:** Are both losses (CE and CAS) in Eq 5 taken into account in the generation of adversarial perturbations with FGSM and PGD?
>
> **A2:** Yes. When evaluated by FGSM and PGD, both of them are taken into account. We have also conducted the ablation study on attacking separate loss terms in Table 4 and 12, where CAS demonstrates a consistent improvement over the baseline defenses.
>
> ---
> **Q3:** In Table 2, what does it mean to have CAS without channel suppression? Is it effectively just a CNN with predictions made from features in different layers?
>
> **A3:** The CAS without channel suppression means the CAS module is inserted, however, the channel suppressing operation (Eq 3) is not applied during either training or testing. In other words, the channels are not reweighted. In this case, the CAS is just a simple auxiliary classifier.
>
> ---
> **Q4:** Do the authors have any intuitions on why having CAS module alone on Block 4 is better than having it on both Block 3 and 4 in general?
>
> **A4:** This is because low-level features cannot produce accurate class predictions, and the low-level channels are weakly correlated with the class. This will lead to less accurate channel importance estimations. When inserted at both Block 3 and Block 4, the inaccurate channel importance estimation will reduce the quality of the inputs to Block 4. Therefore, having the CAS module on Block 4 alone is better.
>
> ---

---

### Public Comment · ~Nicholas_Carlini1 · 2020-11-10
**Understanding the paper claims**

This is an interesting paper. I have some questions about understanding the robustness results.

Looking at Table 2, the claim is that applying CAS to a model trained with adversarial training will increase the robustness from 46.5% accuracy up to 48.88% accuracy, when considering the strongest attacks. Is this +2.38% accuracy on CIFAR-10 the main robustness claim of the paper?

I believe the answer is "no", but to clarify, does the proposed scheme intend to provide any robustness without adversarial training, on just a baseline model?

I'm curious also why performing 100 steps of PGD does *worse* than just 20 steps of PGD in Table 1. Is this to be expected?

---

> ### Author Response · Authors · 2020-11-20
> **Response to Nicholas Carlini**
>
>
> Hi Nicolas,
> Thanks for the comments.
>
> In terms of robustness improvement, yes. However, this is definitely not our only contribution. Our analysis of the channel-wise characteristics of adversarial activations provides unique insights into the robustness of intermediate layer presentations. Our CAS training strategy is simple and flexible, which can be easily applied to robustify the intermediate layers of a DNN. Moreover, our method can also help representation learning and natural training, as shown in Appendix E.
>
> The result for PGD-100 in Table 1 is the Avg-PGD-100 [1] with the margin loss. We have fixed the name in Table 1.
>
> [1] Florian Tramer, Nicholas Carlini, Wieland Brendel, and Aleksander Madry.  On adaptive attacks to adversarial example defenses.arXiv preprint arXiv:2002.08347, 2020.

---

### Public Comment · ~Sebastian_Palacio1 · 2020-11-11
**Choosing the loss function to attack**

As the motivation for CAS comes from analyzing certain properties of the feature space (e.g., magnitude, activation frequency), an evaluation using feature adversaries (Sabour et al., 2016) would provide support to the claim that adversarial attacks show different behavior than clean samples in the feature space (in particular, results from in figures 1 and 2). Moreover, this method was recently used to break many newly proposed defenses with similar motivations (Tramer et al. 2020), and constitutes a strong and simple adaptive attack.

References:
1. Sabour, Sara, et al. "Adversarial manipulation of deep representations." arXiv preprint arXiv:1511.05122 (2015).
2. Tramer, Florian, et al. "On adaptive attacks to adversarial example defenses." arXiv preprint arXiv:2002.08347 (2020).

---

> ### Author Response · Authors · 2020-11-20
> **Response to Sebastian Palacio**
>
> Hi Sebastian,
> Thanks for the suggestion.
> We have conducted the suggested feature attack experiment [1] on CIFAR-10 dataset. We have tested the robustness of ResNet-18 trained using either PGD adversarial training (ADV) or standard training (STD) with or without our CAS module (at Block 4). We used the shared code on GitHub and set the attack step to 100 (according to the paper). The results are reported in the table below. We find that feature attack is not effective against adversarially trained models, as also discussed by the authors [2]. In [2], the difficulty of logit-matching attack when instantiated with the robust classifier was also discussed in Section 6.3. Our defense can improve both the robustness of the standard training and the adversarial training.
>
> Table 1: Robustness of ResNet18 against feature attack on CIFAR-10(%). STD: standard training; STD+CAS: standard training with CAS; ADV: adversarial training; ADV_CAS: adversarial training with CAS. The robustness was computed on the CIFAR-10 test set.
>
> |&nbsp;&nbsp;&nbsp;&nbsp;&nbsp;Attack |&nbsp;&nbsp;&nbsp;STD|&nbsp;&nbsp;STD+CAS  |&nbsp;&nbsp;ADV  |&nbsp;&nbsp;ADV+CAS |
> |:-:|:-:|:-:|:-:|:-:|
> |100-step feature attack|&nbsp;&nbsp;&nbsp;0.00     |&nbsp;&nbsp;3.66     |&nbsp;&nbsp;73.92  |&nbsp;&nbsp;83.63   |

---

### Public Comment · ~Hanshu_YAN2 · 2020-11-16
**How to generate the adv examples during the training and test phases?**

Dear Authors,

For the Eqn (4), actually, this is a negative log-likelihood. Since $\hat p$ is the softmax of (f^{hat} M), the L_CAS is exactly the cross-entropy loss between ($\hat{f} M$) and y.

I also have a question on how the adv examples are generated. During the test phase (Eqn 3), the mask M is selected by the prediction $\hat{y}$. When generating an untargeted attack, the label y is given. So, did you use the $M_{y}$ or $M_{\hat{y}}$?
If you use $M_{\hat{y}}$, does it mean that, for PGD attack, in each iteration, we need to compute the mask $M_{\hat{y}_{iter-index}}$ on the fly?

The last question, which confuses me most, is that, during the test phase, how the gradients of CE loss of PGD attack backward through the CAS module? Getting the index $\hat{y}$ from $\hat{p}$ via \argmax operation is non-differentiable.  I guess, when reweighting the feature maps, one will detach the mask $M_{\hat{y}}$ from the computing graph, is this right?

Thanks

---

> ### Author Response · Authors · 2020-11-20
> **Response to Hanshu Yan**
>
> Hi Hanshu,
> To answer your questions:
> - Yes, Eqn (4) is equivalent to the cross entropy.
> - During the test phase, we use $M_{\hat{y}}$ for both targeted and untargeted attacks ($\hat{y}$). And yes, we need to compute the mask on the fly.
> - Compute the \argmax, reweighting the feature maps,  compute the adversarial loss and backpropagate.

---

### Author Response · Authors · 2020-11-20
**Rebuttal Summary**

We sincerely thank all reviewers for their valuable comments and suggestions. We have made the following updates during the rebuttal.

---
+ Section 3: added detailed explanations on the channel-wise activation frequency.
+ Section 5.1: added benefits of CAS on the representation learning and natural training.
+ Section 5.1: added explanations on the channel suppressing effect.
+ Section 5.1: added explanations and analyses on the robustness of the CAS module.
+ Appendix F.6: added more results (Figure 10) and analyses of channel-wise activation frequency with different thresholds.
+ Appendix A/B/C: reorganized the sections in Appendix.
+ Fixed the typos and added more details in the full text.

---
We have revised our paper according to all the valuable comments and please let us know if there is anything still not clear or any other suggestions.

---

### Decision · Program_Chairs · 2021-01-07
**Final Decision**

**Decision:**

Accept (Spotlight)

**Comment:**

This paper focuses on two new characteristics of adversarial examples from the channel-wise activation perspective, namely the activation magnitudes and the activated channels. The philosophy behind sounds quite interesting to me, namely, suppressing redundant activations from being activated by adversarial perturbations. This philosophy leads to a novel algorithm design I have never seen, i.e., Channel-wise Activation Suppressing (CAS) training strategy.

The clarity and novelty are clearly above the bar of ICLR. While the reviewers had some concerns on the significance, the authors did a particularly good job in their rebuttal. Thus, all of us have agreed to accept this paper for publication! Please carefully address all comments in the final version.